# The Therapeutic Potential of CDK4/6 Inhibitors, Novel Cancer Drugs, in Kidney Diseases

**DOI:** 10.3390/ijms241713558

**Published:** 2023-08-31

**Authors:** Xuan-Bing Liang, Zhi-Cheng Dai, Rong Zou, Ji-Xin Tang, Cui-Wei Yao

**Affiliations:** Guangdong Provincial Key Laboratory of Autophagy and Major Chronic Non-Communicable Diseases, Key Laboratory of Prevention and Management of Chronic Kidney Diseases of Zhanjiang City, Institute of Nephrology, Affiliated Hospital of Guangdong Medical University, Zhanjiang 524001, China

**Keywords:** cell cycle, CDK4/6 inhibitors, cancer, kidney diseases, acute kidney injury, chronic kidney disease

## Abstract

Inflammation is a crucial pathological feature in cancers and kidney diseases, playing a significant role in disease progression. Cyclin-dependent kinases CDK4 and CDK6 not only contribute to cell cycle progression but also participate in cell metabolism, immunogenicity and anti-tumor immune responses. Recently, CDK4/6 inhibitors have gained approval for investigational treatment of breast cancer and various other tumors. Kidney diseases and cancers commonly exhibit characteristic pathological features, such as the involvement of inflammatory cells and persistent chronic inflammation. Remarkably, CDK4/6 inhibitors have demonstrated impressive efficacy in treating non-cancerous conditions, including certain kidney diseases. Current studies have identified the renoprotective effect of CDK4/6 inhibitors, presenting a novel idea and potential direction for treating kidney diseases in the future. In this review, we briefly reviewed the cell cycle in mammals and the role of CDK4/6 in regulating it. We then provided an introduction to CDK4/6 inhibitors and their use in cancer treatment. Additionally, we emphasized the importance of these inhibitors in the treatment of kidney diseases. Collectively, growing evidence demonstrates that targeting CDK4 and CDK6 through CDK4/6 inhibitors might have therapeutic benefits in various cancers and kidney diseases and should be further explored in the future.

## 1. Introduction

The dysregulation of the cell cycle is closely associated with the development of various diseases, particularly cancer, where abnormal cell cycle activation is a hallmark of carcinogenesis. Moreover, dysregulated cell proliferation is also implicated in the pathogenesis of renal diseases, cardiovascular diseases and nervous system disorders. Consequently, there has been a considerable focus on targeting pathways that regulate the cell cycle across different disease disciplines for several years.

CDK4/6 inhibitors are a novel class of therapy that selectively targets the CDK4/6-cyclin D complex to regulate the cell cycle. In recent years, the FDA has approved several CDK4/6 inhibitors, such as palbociclib, ribociclib and abemaciclib, for the treatment of HR-positive breast cancer patients. These inhibitors have shown remarkable results and have significantly improved the options for breast cancer treatment [1,2,3,4]. The efficacy of CDK4/6 inhibitors is now being explored in other diseases, as it has been discovered that CDK4/6 may play a role in pathological conditions beyond cancer. Preliminary studies have already demonstrated a potential renal protective effect of CDK4/6 inhibitors. As research progresses, it is hoped that CDK4/6 inhibitors can be extended to treat a broader range of diseases, providing new therapeutic options for patients.

This review provides an overview of the cell cycle in eukaryotes and emphasizes the essential role of CDK4/6 in regulating this process. The application of CDK4/6 inhibitors in cancer treatment is discussed, with a particular focus on their potential in treating kidney diseases. Additionally, the review explores the various therapeutic strategies and challenges associated with using CDK4/6 inhibitors for treating kidney diseases. A deeper understanding of the mechanism and role of CDK4/6 inhibitors in kidney disease will aid in the development of more effective drugs for the treatment of these conditions.

## 2. Cell Cycle and Its Regulation

Cell division is a crucial process in biology that is tightly regulated by conserved signaling pathways, ensuring the accurate replication of DNA. In mammals, the cell cycle consists of four phases: G1, S, G2 and M, with an additional phase called G0, where cells exit the cycle and enter a quiescent state [5,6]. In the kidney, most cells are arrested in the G0 phase, but certain highly specialized cells, like neurons and cardiomyocytes, permanently withdraw from the cell cycle. The progression of the cell cycle is controlled by key components such as cyclins, cyclin-dependent kinases (CDKs) and cyclin-dependent kinase inhibitors (CKIs), which interact to form a precise regulatory network [7].

The discovery of CDK, a key regulator of cell cycle progression, dates back to the 1980s when it was first identified in sea urchin eggs [8]. The CDK family comprises serine/threonine kinases and can be categorized into subfamilies based on their evolutionary relationships [9,10]. Some CDKs, such as CDKs 1–6, 11 and 14–18, play a crucial role in regulating cell cycles, while others, including CDKs 7–13, 19 and 20, participate in transcription processes. Despite their structural similarities, each CDK has a specific function that is regulated by context-specific cyclins. The activation of CDKs is essential for the proper functioning of cell cycle proteins, and reciprocally, their interaction with these proteins significantly affects CDK activity. In addition to CDK-cyclins complexes, cell cycle protein-dependent protein kinase inhibitors (CKI) also contribute to the regulation of cell cycle progression. The CDK inhibitors consist of two main families: the INK4 family, which includes p16^INK4a^ (Cdkn2a), p15^INK4b^ (Cdkn2b), p18^INK4c^ (Cdkn2c) and p19^INK4d^ (Cdkn2d), specifically targeting CDK4/6-cyclin D complexes; and the Cip/Kip family members, including p21^Cip1^ (Cdkn1a), p27^Kip1^ (Cdkn1b) and p57^Kip2^ (Cdkn1c), which broadly inhibit CDK-cyclins complexes. These inhibitors play a crucial role in regulating cell cycle progression by disrupting the activity of CDKs and their associated cyclins.

### 2.1. Classical Regulation of G1/S Phase Transition

According to the classical regulatory pathway of cellular G1 to S phase transition, D-type cyclins are the regulators of CDK4/6, and their expression is increased in the early G1 phase in response to various mitotic stimuli [11]. The regulation of CDK4/6 activity is crucial for the inactivation of the retinoblastoma protein (Rb). When CDK4/6 forms a heterodimer with cyclin D, CDK4 and CDK6 are activated. Subsequently, the dimeric complex phosphorylates Rb1 and other Rb1-like “pocket” proteins, namely P130 and P107, which are respectively referred to as retinoblastoma-like proteins 1 and 2, at multiple sites [12,13,14]. In its hypophosphorylated state, Rb1 binds to the trans-activation domain of the E2F transcription factor family of proteins, thereby inhibiting the transcription of genes necessary for cell cycle progression [15,16].

Upon activation, CDK4/6 can phosphorylate Rb1, thereby relieving the repression of RB1 by E2F transcription factors and facilitating the dissociation of E2F from Rb1 [17,18]. The expression of genes essential for the transition to the S phase relies on E2F, which promotes DNA synthesis by activating various genes and facilitating the transcription of E-type proteins. Moreover, cyclin E plays a crucial role in the late G1 phase by activating CDK2 to form the CDK2-cyclin E complex. This complex then phosphorylates Rb1, thus releasing the inhibition of E2F and promoting cellular entry into the S phase [19,20,21,22,23] (Figure 1). Conversely, during the S-phase, CDK2 forms a complex with cell cycle protein A, while Rb1 undergoes transitional phosphorylation, thereby mediating the transcriptional control of DNA synthesis. Additionally, the CDK1-cyclin B complex regulates the progression through the M phase.

### 2.2. Non-Classical Regulation of G1/S Phase Transition

The classical understanding of G1/S transition is widely accepted, but the exact role of certain CDKs in this process may be more complex. Experiments using knockout mouse models for CDK4 and CDK6 challenge the traditional view of G1/S transition, which suggests a reliance on human cellular CDKs. Interestingly, certain cell types can enter the S phase without the presence of CDK 4 and CDK 6. For instance, mice can survive with a deficiency in either CDK4 or CDK6 alone. CDK4 deficiency typically leads to stunted growth and compromised reproductive and endocrine function [24,25,26], while CDK6 deficiency primarily affects the hematopoietic system, resulting in the reduced thymus and spleen cell function and a decreased number of peripheral blood erythrocytes [27]. Simultaneous knockout of CDK4 and CDK6, although causing embryonic mouse death, is considered to be a result of erythrocyte defects caused by CDK6 deficiency, leading to severe anemia [27].

However, in vitro experiments have shown that mouse embryonic fibroblasts (MEFs) with mutations in both CDK4 and CDK6 can still undergo cell cycle progression from G1 to S phase when stimulated by serum. This suggests that CDK4/6 is not necessarily required for the stimulation of cell proliferation in quiescent cells. Instead, it is believed that the phosphorylation of RB by an atypical CDK2-cyclin D complex may play a role in this process [27,28]. The discovery of these non-classical pathways challenges the traditional understanding of the G1 to S phase transition in the cell cycle, which has mainly focused on CDK4/6. Non-classical models propose that CDK2 can directly interact with cyclin E and cyclin D, forming a complex without requiring CDK4/6 activation. This complex then facilitates the cell cycle transition from the G1 to S phase [28,29,30] (Figure 2). However, the exact mechanisms underlying this process are still not fully understood.

In order to maintain the accuracy of DNA replication, mammals have developed a sophisticated quality control system comprising four distinct cell cycle checkpoints. These checkpoints play a vital role in ensuring the smooth progress of the cell cycle and regulating it with precision in eukaryotic cells. The transition from the G1 phase to the S phase is carefully controlled by the G1/S checkpoint. This initial checkpoint is responsible for detecting any DNA damage, halting the cell cycle for necessary DNA repair, and safeguarding the integrity of the genome [31]. The G1 phase represents a critical period during which cellular DNA begins replication after passing through the G1/S checkpoint and entering the S phase.

In addition, the S checkpoint follows, which is triggered by either DNA that has escaped the previous checkpoint or damaged DNA, effectively halting the progression of the cell cycle into the S phase [31]. The third checkpoint, referred to as the G2/M checkpoint, ensures the successful completion of mitosis [31]. The fourth checkpoint, known as the intermediate or spindle checkpoint, plays a crucial role in monitoring the cell cycle progression [31]. These checkpoints work together to establish a highly intricate and interconnected network, serving as a rigorous quality control mechanism that continuously scrutinizes and regulates the entire advancement of the cell cycle.

## 3. CDK Inhibitors Have Emerged as Key Players in the Field of Cancer Therapeutics

CDK inhibitors are pharmacological agents that specifically target abnormal CDK activity in malignant cells. However, the first generation of clinically developed inhibitors faced limitations due to the need for selectivity, as CDKs have structural similarities [32,33]. An example of such an inhibitor is Flavopiridol, which targets multiple CDKs, including CDK1, CDK2, CDK7 and CDK9, along with CDK4/6. Unfortunately, clinical trials with Flavopiridol have shown disappointing results in patients with hematologic malignancies and solid tumors. Furthermore, studies on mouse embryos have indicated that the absence of CDK1 can impair development, suggesting that non-specific CDK inhibitors can have toxic effects on cellular growth and development [34]. CDK7, another target of these inhibitors, is also involved in cell cycle transition, but its exact function is not fully understood. Therefore, broad-spectrum CDK inhibitors with low selectivity often have toxic side effects, making it challenging to find safe and specific inhibitors.

Similarly, the highly sought-after drug roscovitine has proven ineffective in tests involving patients with triple-negative breast cancer, non-small cell lung cancer and other advanced solid tumors due to its limited effectiveness and toxicity [9]. Second-generation pan-CDK inhibitors, such as dinaciclib and SNS-032, have improved selectivity towards a lesser number of CDKs and decreased toxicity profiles. However, they still exhibit lower levels of inhibitory activity against CDK4/6 [33]. Given the underwhelming success of these first- and second-generation inhibitors, it is crucial to discover inhibitors that provide greater selectivity in targeting specific CDKs. Some progress has been made in targeting non-cell cycle CDKs, including CDK9 and CDK12 [35,36]. Nevertheless, the inhibitors that have made the most advancements and are currently in clinical use primarily focus on CDK4/6.

These inhibitors have predominantly been primarily developed through chemical screening and optimization, involving the incorporation of pyrido[2,3-d] pyrimidin-7-one compounds with a 2-amino pyridine side chain at the C2 position [37]. Consequently, CDK4/6 inhibitors such as palbociclib, ribociclib and abemaciclib, with potent efficacy and reduced toxicity, have quickly transitioned from research laboratories to clinical trials. In fact, all three of these drugs have gained FDA approval for the treatment of metastatic breast cancer. Compared to previous CDK inhibitors, these compounds exhibit high selectivity in inhibiting CDK4/6. They have more than a 100-fold affinity for CDK4 and CDK6 relative to other CDKs [38]. All three CDK4/6 inhibitors are administered orally, with palbociclib being the first to demonstrate clinical efficacy and extensively investigated in breast cancer treatment. Ribociclib bears a close structural resemblance to palbociclib, whereas abemaciclib deviates significantly in structure from the first two. Additionally, abemaciclib exhibits lower selectivity compared to palbociclib and ribociclib. However, it offers potential applications in central nervous system disorders such as brain cancer and secondary brain metastases due to its enhanced ability to crosse the blood–brain barrier more effectively [39].

## 4. CKD4/6 Inhibitors in the Treatment of ER-Positive Breast Cancer

ER-positive luminal breast cancer serves as the ideal model for investigating the effectiveness of CDK4/6 inhibitors. These inhibitors target the reliance of these cancer types on cyclin D1 for the initiation of the G1-to-S-phase transition. Previous clinical trials have established that endocrine anti-estrogen therapy is the cornerstone of systemic treatment for ER-positive breast cancer. However, resistance to endocrine therapy often arises when it fails to adequately control tumor progression. Various mechanisms have been proposed as contributors to endocrine resistance, including loss or mutations of the ER, alterations in ER pathways, dysregulation of cell cycle signaling molecules and activation of escape pathways [40,41]. The addition of CDK4/6 inhibitors to anti-estrogen therapy has been shown to improve progression-free survival (PFS), underscoring their potential as targeted and effective treatments for ER-positive breast cancer [42,43,44,45].

Currently, the FDA has approved the inhibition of CDK4/6 as a novel therapeutic approach for HR-positive breast cancer. Three CDK4/6 inhibitors, when used in combination with endocrine therapies, have shown significant efficacy and have partially overcome the limitations posed by endocrine resistance. Studies suggest that Palbociclib, a CDK4/6 inhibitor, can induce growth arrest in breast cancer cells that have developed acquired resistance to endocrine therapy [46]. The selection of the appropriate patient population for CDK4/6 inhibition in cancer therapy depends on the presence of functional Rb within the CDK4/6 axis. In most cases, tumors that retain functional Rb proteins, which are the primary targets of CDK4/6 inhibitors, during the development of endocrine resistance are suitable for CDK4/6 inhibition [47]. Understanding the exact pharmacological mechanism of action of CDK4/6 inhibitors is crucial to further research in this field.

When considering the differences in Rb-pathway behavior between ER-positive and ER-negative diseases, it is important to note that in ER-positive breast cancer, various factors contribute to the synergistic effect observed. Evidence suggests that ER-positive breast cancers typically maintain an Rb expression signature, which may play a role in this synergy [48]. Furthermore, activation of upstream signaling pathways such as RAS, mitogen-activated protein kinase (MAPK) and mammalian target of rapamycin (mTOR), and nuclear receptors (such as estrogen receptor (ER) in the mammary epithelium, regulates the cell cycle progression by promoting the formation of the CDK4/6-cyclin D complex. This, in turn, leads to uncontrolled cell proliferation [49,50,51,52,53,54]. CDK4/6 serves as a critical regulator of the cell cycle by forming a complex with cyclin D. This complex directly phosphorylates the RB, leading to the release of the transcription factor E2F and facilitating the transcription of genes involved in cell cycle progression [55]. Consequently, this process allows the cell cycle to advance from the G1 phase to the S phase, enabling DNA replication [56]. Furthermore, the precise pharmacological mechanisms of action of CDK4/6 inhibitors have been extensively investigated. Multiple explanations have been proposed regarding how these inhibitors function. The traditional model suggests that CDK4/6 inhibitors primarily block the activity of these kinases, particularly in ER-positive breast cancer (Figure 3a). Additionally, it has been postulated that CDK4/6 inhibitors indirectly inhibit the activity of CDK2, thereby impeding the progression from G1 to S phase [30,57,58] (Figure 3b,c). These findings emphasize the crucial role of CDK4/6 inhibitors in inhibiting the progression of ER-positive breast cancer.

However, despite the benefits of CDK4/6 inhibitors in controlling HR-positive breast cancer, not all patients experience a positive response to these drugs. This is because CDK4/6 inhibitors typically work downstream of endocrine therapy, and resistance can develop to both forms of treatment. This resistance may be caused by various factors, such as mutation or deletion of Rb [28,29]; overexpression of p16 in the presence of functional Rb [59]; amplification of CDK4, CDK6 [60,61,62], or CDK2 in the CDK2-cyclin E axis [28,63]; overexpression of E2F; and other dysregulations in the cell cycle and bypass mechanisms. Consequently, many patients who initially respond to CDK4/6 inhibitors eventually develop acquired drug resistance [64]. However, there is potential for improving treatment outcomes by combining CDK4/6 inhibitors with mTOR and other kinase and checkpoint inhibitors [65,66,67,68,69,70,71]. Hence, investigating and exploring the optimal combination regimen of CDK4/6 inhibitors with other endocrine therapies is crucial for enhancing drug sensitivity.

## 5. CDK 4/6 Inhibitors in the Treatment of Renal Diseases

In addition to their role in cancer treatment, CDK4/6 inhibitors have demonstrated beneficial effects in non-cancerous diseases as well. The use of CDK inhibitors in treating kidney disease was first reported in 1997, where the inhibition of the cell cycle showed improvement in embolic proliferative glomerulonephritis [72]. Subsequently, the application of CDK inhibitors expanded to other renal diseases, highlighting the antiproliferative effects on podocytes in crescentic glomerulonephritis. However, most of the research in this area has focused on CDK2 inhibitors [73]. As investigations progressed, it became evident that CDK4/6 inhibitors also exhibit promising efficacy in the treatment of renal disease. This review aims to delve into the mechanisms underlying the therapeutic effects of CDK4/6 inhibitors in various renal diseases.

### 5.1. CDK4/6 Inhibitors and Acute Kidney Injury

Acute kidney injury (AKI) is a clinical condition characterized by a rapid decline in renal function, usually caused by acute ischemia, infection, or drug toxicity-induced death of renal tubular cells [74,75,76,77,78,79]. Under normal physiological conditions, renal tubular epithelial cells (RTECs) remain mostly quiescent. However, they possess a robust regenerative capacity, and their self-proliferation plays a crucial role in kidney injury repair, which involves the cell cycle [22]. Among the RTECs, the proximal S3 segment is particularly vulnerable to injury following acute ischemic or toxic damage. Previous studies have established the significant contribution of these cells in the development of kidney injury diseases [80,81,82,83,84,85,86]. This proliferative response is believed to play a critical role in kidney repair and regeneration [86,87]. Moreover, studies have demonstrated a protective effect of blocking or delaying cell cycle entry during AKI [88]. In fact, AKI often induces DNA damage, which activates ataxia telangiectasia mutated (ATM) or ataxia telangiectasia and Rad3-related (ATR) proteins. These proteins phosphorylate downstream targets, including p53 and checkpoint kinase 2 (CHK2), which belong to the phosphatidylinositol 3-kinase family. Consequently, p21, a CIP/KIP family cell cycle inhibitor, is generated, causing tubular epithelial cells to arrest in the G1 or G2/M phase [89,90,91]. p21 has been observed to be rapidly induced in proximal tubular cells in various experimental models of AKI, and its involvement in the assembly of the CDK4-cyclin D complex and its cytostatic effects have been identified [92,93,94]. Therefore, it is speculated that the involvement of p21/p27 in G1/S transitions serves as an adaptive mechanism. Moreover, it has been demonstrated that the depletion of p21 exacerbates cisplatin-induced acute kidney injury [94]. Several studies have indicated that enhancing p21 expression and reducing CDK2 levels may induce G1 arrest, thereby alleviating AKI by preventing apoptosis in proximal tubular cells [85,88,95,96,97]. However, the clinical utilization of CDK2 inhibitors is limited due to their toxic side effects, which can also impede other CDKs involved in DNA transcription and metabolic processes, resulting in G2/M arrest and S phase arrest, respectively. This can promote apoptosis and fibrosis [83,85,88,98]. The introduction of CDK4/6 inhibitors has shown promising renal protective effects with fewer adverse effects and improved inhibition of G1/S transition [33,99,100]. The presence of functional Rb1 in RTECs is believed to be associated with the ability of hypophosphorylated Rb to block cell cycle progression from G1 to S phase [99,101]. A mouse study investigated the impact of ribociclib on cisplatin-induced nephrotoxicity and suggested that it potentially mitigates renal damage at the early stages through an Rb1-dependent mechanism [99]. Administration of ribociclib resulted in reduced expression of phosphorylated Rb and displayed a protective effect against cisplatin-induced AKI, leading to improved renal function, decreased tubular injury and reduced levels of active/cleaved cysteines [99]. In vivo and in vitro experiments utilizing CDK4/6 inhibitors temporarily halted the traversal of the S phase, with cells re-entering the cell cycle after treatment. These inhibitors have demonstrated beneficial effects, including a reduction in the expression of inflammatory markers such as TNF and MCP-27, a significant decrease in macrophage infiltration, and a reduction in serum creatinine and urea nitrogen levels [102]. Additionally, another study provides strong evidence supporting the inhibition of CDK4/6 in preventing cell-cycle progression. Palbociclib and ribociclib have shown unique and potentially advantageous inhibitory activity on organic cation transporter-2 (OCT-2), in addition to their targeted inhibition of CDK4/6 activity [103] (Figure 4). Furthermore, an animal experiment confirmed that pretreatment with ribociclib in septic mice alleviated sepsis-induced AKI. The study also suggested the involvement of the mTOR/AKT pathway in the renal protective effect of ribociclib [104]. However, a study that examined the clinical features and histopathology associated with CDK4/6 inhibitors reported adverse events in patients with biopsy-confirmed AKI [105]. Clinical trials investigating CDK4/6 inhibitors have also reported elevated serum creatinine levels in some patients, though it remains difficult to determine whether the drug is the actual cause [105,106]. Elevated serum creatinine levels are believed to be the result of reversible inhibition of renal efflux transport proteins rather than acute kidney injury [106].

Further evidence supports the notion that reversible increases in serum creatinine, observed after treatment with abemaciclib, may be attributed to inhibition of renal transporters such as organic cation transporter-2 (OCT-2), multidrug, toxin extrusion-1 (MATE-1) and MATE2-K. Importantly, these increases do not affect the glomerular filtration rate [103,107] (Figure 4). The exact mechanism by which CDK4/6 inhibitors exert their renal protective effects remains unknown. Despite the potential for reversible creatinine elevation, the favorable pharmacological characteristics observed in numerous trials make CDK4/6 inhibitors promising candidates for AKI prevention.

### 5.2. CDK4/6 Inhibitors and Chronic Kidney Disease

Chronic kidney disease (CKD) is a long-term condition that occurs when there is a structural or functional abnormality in the kidneys lasting for more than three months. It has a detrimental impact on overall health, affecting multiple organ systems, especially the cardiovascular system and mineral–bone metabolism. This leads to various complications that are associated with increased morbidity and mortality, placing a significant socioeconomic burden on society [108,109]. Numerous epidemiological studies have demonstrated a close link between AKI and CKD. Those who survive AKI often go on to develop CKD, indicating a bidirectional relationship between these two conditions [110,111,112,113].

CKD is characterized by the presence of tubulointerstitial fibrosis (TIF), tubular atrophy, and the accumulation of extracellular matrix proteins. While increased cell cycle processes play a role in promoting repair after acute injury, dysregulation of renal tubular epithelial cells during this process can contribute to fibrosis [114,115,116]. Proximal tubular cells (PTCs) are the predominant cell type in the kidney and are crucial sites of injury. Therefore, much research has focused on understanding the transition from AKI to CKD, with particular emphasis on PTCs and resident fibroblasts, which are responsible for producing extracellular matrix proteins [91,114,117,118]. However, the exact mechanistic role of the cell cycle in CKD is still being investigated. Various studies have aimed to unravel the process of maladaptive repair that ultimately leads to the development of progressive fibrotic nephropathy [119,120].

The results of the two-week intervention with palbociclib in the UniNx/AngII and adenosine nephropathy CKD models suggest that palbociclib can effectively delay the progression of CKD. This is achieved by reducing proximal tubular cell death and inhibiting cell cycle progression in the G1/S phase. As a result, renal function is improved, and tubular injury and fibrosis are reduced. These beneficial effects may be mediated through the STAT3/IL-1β pathway, as illustrated in Figure 5 [121] (Figure 5). Previous studies have provided some evidence indicating that CDK4/6 might induce STAT3 activation through mechanisms involving the methyltransferase EZH3 or direct binding to the STAT2 promoter [122,123]. Additionally, it has been demonstrated that STAT3 plays a significant role in renal tubular interstitial fibrosis and becomes activated in response to injured renal tubules [124,125]. However, the exact mechanism by which CDK4/6 triggers STAT3 activation remains unclear. Additionally, further experiments have indicated that the deletion of the cyclin D1 actually worsens chronic kidney injury. This suggests that strategies aimed at preventing chronic kidney disease should prioritize the inhibition of CDK4/6 rather than cyclin D1 [121]. The observation that decreased expression of the CDK4/6 inhibitor p15 is closely associated with reduced eGFR suggests a causal relationship [121]. However, more information regarding the involvement of cell cycle processes in the G1/S transition of chronically injured tubules still needs to be provided.

According to the classical pathway, CDK4/6 binds to the cyclin D1 and phosphorylates Rb, leading to the release of E2F and promoting G1/S progression. It has been observed that palbociclib, a CDK4/6 inhibitor, cannot provide protection against the survival of proximal renal tubules in the absence of Rb. Retarding G1/S cell cycle progression has been shown to have a protective effect. Inhibition of CDK4/6 reduces the number of cells progressing to the S phase, which in turn decreases the population that undergoes G2/M arrest. This shift is associated with a more pro-fibrotic phenotype characterized by increased expression of TGF-β and CTGF [126,127]. In vitro studies have demonstrated that high concentrations (10 μmol/L) of palbociclib significantly impede the proliferative activity of human renal tubular epithelial cells (HK-2 cells). This inhibition is accompanied by an increase in the proportion of cells in the G0/G1 phase and upregulation of p16, p21 and p53 protein levels [127]. Senescent cells can secrete a variety of bioactive molecules known as senescence-associated secretory phenotypes (SASP), which can trigger chronic inflammatory responses and fibrosis. This process is believed to be a key mechanism behind aging-related organ fibrosis [128,129,130]. Mechanistic studies have shown that the increased severity of AKI in mice lacking the Smad7 gene is associated with heightened activation of TGF β/Smad3-p21 signaling [131]. This activation leads to sustained G1 cell cycle arrest and subsequent development of fibrosis. Contrary to previous findings suggesting that CDK4/6 inhibitors protect renal function, palbociclib may actually promote renal fibrosis by inducing cellular senescence in renal cells. While cell cycle arrest can be beneficial for repairing DNA damage or preventing abnormal cell divisions, prolonged arrest can contribute to the acquisition of a pro-fibrotic phenotype if PTCs are unable to re-enter the cell cycle [126]. Contrary to previous findings suggesting that CDK4/6 inhibitors protect renal function, palbociclib may actually promote renal fibrosis by inducing cellular senescence in renal cells. Therefore, the duration of CDK4/6 inhibition is crucial, as transient exposure to cell cycle inhibitors may have a protective effect. However, prolonged exposure could potentially lead to AKI or contribute to the progression of AKI to CKD [105]. Consequently, when using palbociclib in cancer patients with concomitant CKD, careful consideration should be given to the risk of progressive renal fibrosis and declining renal function.

### 5.3. CDK4/6 Inhibitors and Polycystic Kidney Disease

Dysregulation of CDK4 and CDK6 is highly prevalent in various human diseases, especially cancer. Inhibitors that target these kinases are currently being used to control the growth of tumors [100]. Previous studies have primarily focused on the Rb protein family, including Rb, P107 and P130, as the main targets of CDK4/6 [132]. However, in the case of polycystic kidney disease (PKD), researchers are exploring different substrates that could serve as new targets for CDK4/6 inhibitors. PKD is a common genetic disorder caused by mutations in the PKD1 and PKD2 genes, which encode polycystin-1 (PC-1) and polycystin-2 (PC-2). Autosomal dominant polycystic kidney disease (ADPKD) is the most common form of PKD [133,134,135]. Cysts form due to the excessive proliferation of the epithelial lining in the collecting ducts or renal tubules, which is a distinct pathological feature of ADPKD [134,136]. While the expansion rate of cysts in ADPKD is generally slower compared to tumor proliferation, the signaling and process alterations observed in ADPKD share similarities with those seen in cancer, such as dysregulation of the cell cycle [137]. SMYD2, a novel substrate of CDK4/6, is found to be overexpressed in renal tissues and cell lines derived from mice with ADPK [138]. Both CDK4/6 and SMYD2 dysregulation have also been reported in breast cancer [139]. Previous research suggests that the interaction between CDK4/6 and Rb occurs through phosphorylation, whereas SMYD2 interacts with Rb and methylates it. Moreover, it has been proposed that the binding of SMYD2 to CDK4/6 necessitates the presence of the (S)ET structural domain [140]. These findings hint at an inherent association between CDK4/6 and SMYD2, either directly or indirectly through Rb. The interaction between CDK4/6 and SMYD2 leads to the subsequent phosphorylation and activation of SMYD2, resulting in SMYD2-mediated modifications of histones, including methylation of histone H3 lysine 4 (H3K4) and histone H3 lysine 36 (H3K36), which in turn influences gene transcription [140] (Figure 6). However, there is limited research on the application of CDK4/6 inhibitors for the treatment of PKD. Targeting the overexpression of CDK4/6 and SMYD2 with inhibitors has shown promising results in partially restoring primary cilia in both tumor and cystic cells [140]. These findings further support the therapeutic potential of CDK4/6 inhibitors, either alone or in combination with other inhibitors, for the treatment of PKD.

### 5.4. CDK4/6 Inhibitors and Nephritis

Since CDK2-cyclin E is known to play a crucial role in the cell cycle progression from the G1 to S phase, most previous studies investigating cell cycle inhibition in nephritis have focused on CDK2 inhibitors [141]. Tethered cell proliferation in Thy1 glomerulonephritis has been found to be associated with increased expression and activity of the CDK2-cyclin A complex [142]. The initiation of mesangial cell proliferation is linked to reduced levels of p27^Kip1^ during the peak of mesangial cell proliferation [142]. In a study conducted on rats with Thy1 glomerulonephritis, treatment with roscovitine showed beneficial effects on renal function, including increased creatinine clearance, elevated urinary excretion, decreased proteinuria and reduced hematuria [143].

In mice with systemic lupus erythematosus (SLE), the combination of seliciclib and low-dose methylprednisolone, administered after the onset of the disease, has shown increased efficacy in prolonging lifespan and reducing proteinuria and renal damage compared to treatment with either agent alone [144]. As a result, CDK2 inhibitors may hold promise as a practical therapeutic approach for treating diseases characterized by excessive cell proliferation in experimental glomerulonephritis [72]. However, their clinical use has been limited due to toxicity, lack of clinical activity and pharmacokinetic issues. CDK2 inhibitors have not been utilized in clinical studies. Consequently, there is growing interest in the development of inhibitors that target CDK4/6 with fewer side effects.

Treatment with the CDK4/6 inhibitor palbociclib has shown promise in reducing inflammation in the facial skin and lymph nodes of lupus-susceptible MRL-LPR female mice, leading to a decrease in inflammation [145]. However, a study examining the effects of palbociclib on bleomycin-induced pulmonary fibrosis found elevated levels of inflammatory cells in mice after treatment with the inhibitor [146]. Moreover, there have been reports of adverse reactions to palbociclib treatment, such as a patient with a history of psoriatic arthritis developing Stevens–Johnson syndrome-like skin lesions [147]. Additionally, progressive clinical case reports have indicated an association between the use of CDK4/6 inhibitors and Subacute Cutaneous Lupus Erythematosus (SCLE) [148,149,150]. Thus, while CDK4/6 inhibitors show potential for treating lupus and other conditions, their use may be accompanied by certain risks and side effects.

The notable characteristic of these skin lesions is the increased inflammation observed following palbociclib treatment, which contradicts previous findings. Initially, it was hypothesized that CDK4/6 inhibitors, like Palbociclib, would inhibit neutrophil activation [151]. However, the impact of palbociclib on the activation of various inflammatory cell types remains uncertain, and there is limited information regarding the role of CDK4/6 inhibitors in the development of nephritis, with the underlying mechanisms still unclear. Therefore, a more thorough examination is needed to carefully evaluate the benefits and drawbacks of CDK4/6 inhibitors for the treatment of nephritis.

## 6. Conclusions

CDK4/6 inhibitors have emerged as a promising class of cell cycle-regulating drugs that have demonstrated significant efficacy in the treatment of breast cancer and are currently undergoing preclinical and clinical trials. The therapeutic benefits of CDK4/6 inhibitors extend beyond their use in cancer treatment. In this article, we provide a comprehensive overview of the pivotal role of CDK4/6 in cell cycle processes, review the therapeutic potential of CDK4/6 inhibitors in cancer management, and explore the potential application and the possible side effects of these inhibitors in the treatment of kidney diseases. However, the utilization of CDK4/6 inhibitors for the treatment of renal diseases is currently underexplored, with existing research primarily limited to animal or in vitro cellular models. This highlights the necessity for additional clinical studies to further investigate their potential in this area. The precise mechanisms and specific roles of CDK4/6 inhibitors in treating various renal diseases, including AKI, CKD, PKD and other related nephritis, are still largely unknown. Future research should prioritize investigating the role and mechanism of CDK4/6 inhibitors in treating kidney diseases by utilizing animal models and conducting in vitro cell experiments. Furthermore, additional clinical studies are necessary to validate their effectiveness in various kidney diseases and assess any potential side effects. Combining CDK4/6 inhibitors with other endocrine therapies has demonstrated improved drug sensitivity and enhanced therapeutic outcomes. Therefore, further exploration is needed to identify optimal combination regimens. CDK4/6 inhibitors are anticipated to have broad applications in the treatment of not only various cancers but also kidney diseases. Therefore, further exploration is needed to identify optimal combination regimens. CDK4/6 inhibitors are anticipated to have broad applications in the treatment of not only various cancers but also kidney diseases.

## Figures and Tables

**Figure 1 ijms-24-13558-f001:**
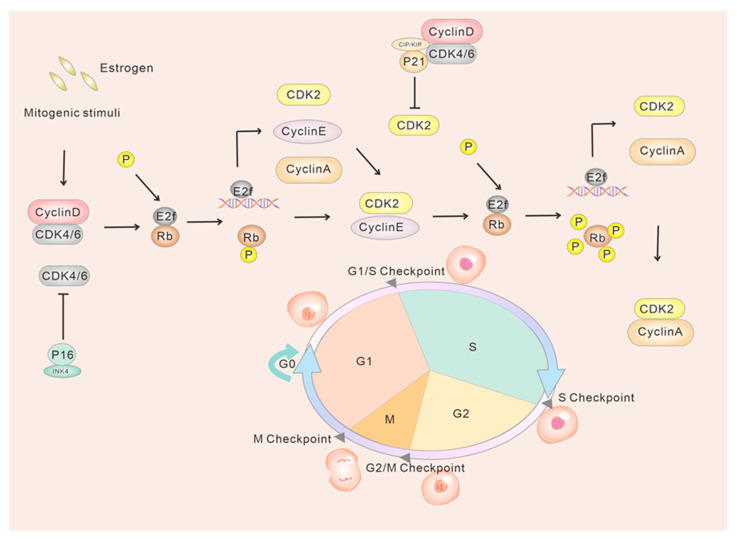
Model diagram of CDK4/6 and G1/S phase transition. Quiescent cells in the G0 or early G1 phase exhibit hypophosphorylated retinoblastoma protein (RB1), which inhibits the transcriptional activity of the E2F. The INK4 protein p16 acts as a suppressor of CDK4/6 activation. The mitogenic and estrogen receptor signaling upregulates the transcription of cyclin D. The cyclin D forms a complex with CDK4/6 to phosphorylate RB, partially activating the E2F-family proteins and leading to the transcription of cyclin A and cyclin E, as well as CDK2. RB phosphorylation also induces chromatin remodeling that promotes transcription. CDK4/6–cyclin D complexes sequester CIP/KIP proteins, diminishing their inhibitory effect on CDK2 and lowering the threshold for CDK2 activation by cyclin E. With increasing cyclin E levels, cyclin E binds with CDK2 to hyperphosphorylate RB, establishing a positive feedback loop through E2F. This loop releases and fully activates E2F, driving the cell from G1 to S phase.

**Figure 2 ijms-24-13558-f002:**
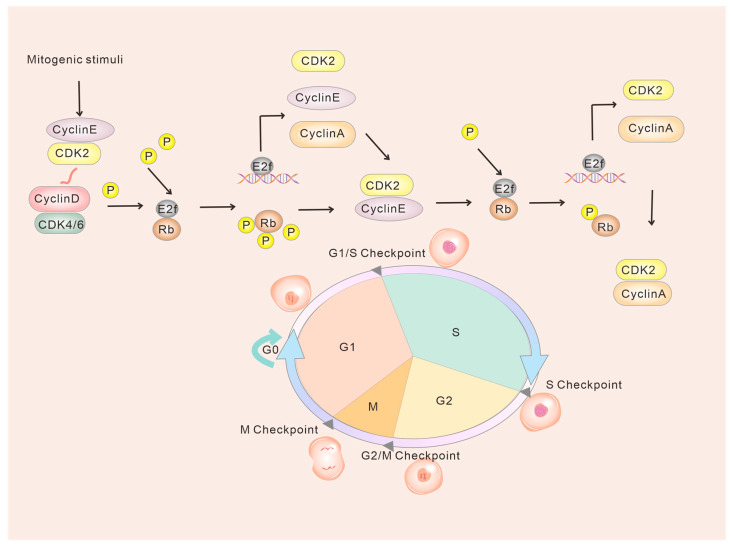
Model diagram of non-classical CDK4/6 and G1/S transition. CDK2 is active during early G1 through direct formation of complexes with cyclins E and potentially cyclin D. Both CDK4/6 and CDK2 phosphorylate RB and drive the G1-S phase transition.

**Figure 3 ijms-24-13558-f003:**
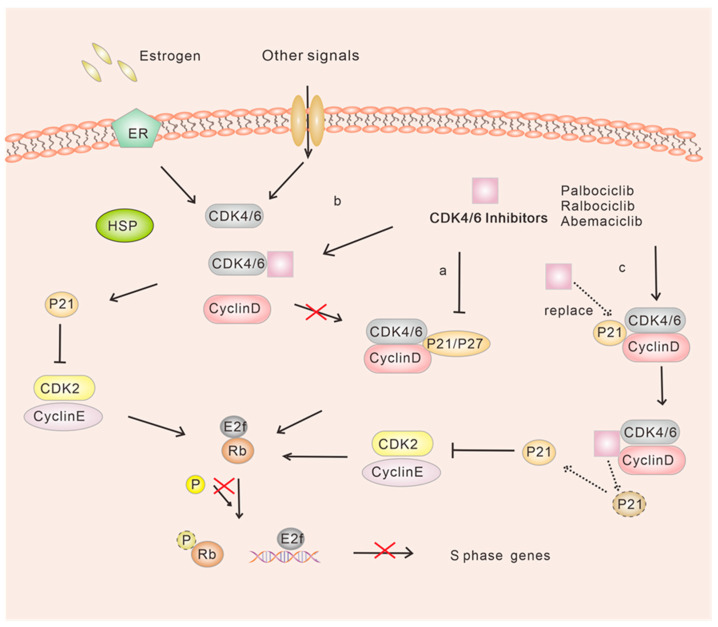
Model diagram of G1/S cyclin-dependent kinase inhibition by CDK4/6 inhibitors. a. CDK4/6 inhibitors inhibit active CDK4/6-cyclin D-p21/p27 allosteric enzymes and block RB phosphorylation of CDK4/6; b. CDK4/6 inhibitors bind to monomeric CDK4/6 and prevent the formation of CDK4/6-cyclin D-p21/p27 trimer. Free p21 then binds to and inhibits the cyclin E/CDK2, preventing phosphorylation of RB; c. CDK4/6 inhibitors directly inhibit the catalytic activity of CDK4/6, which then displaces p21, freeing it to inhibit cyclin E/CDK2 further.

**Figure 4 ijms-24-13558-f004:**
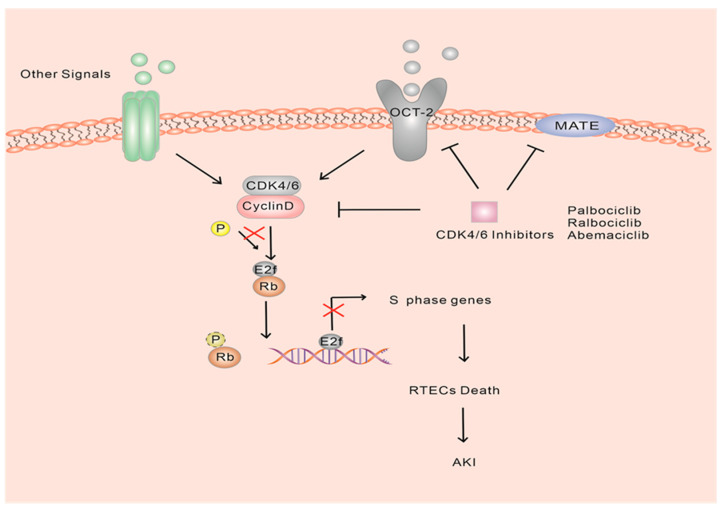
Model diagram of the effect of CDK4/6 inhibitors on AKI. In acute kidney injury, CDK4/6 inhibitors may delay G1 entry to the S phase through different pathways by inhibiting CDK4/6 activity and other substrates, such as OCT-2 and MATE.

**Figure 5 ijms-24-13558-f005:**
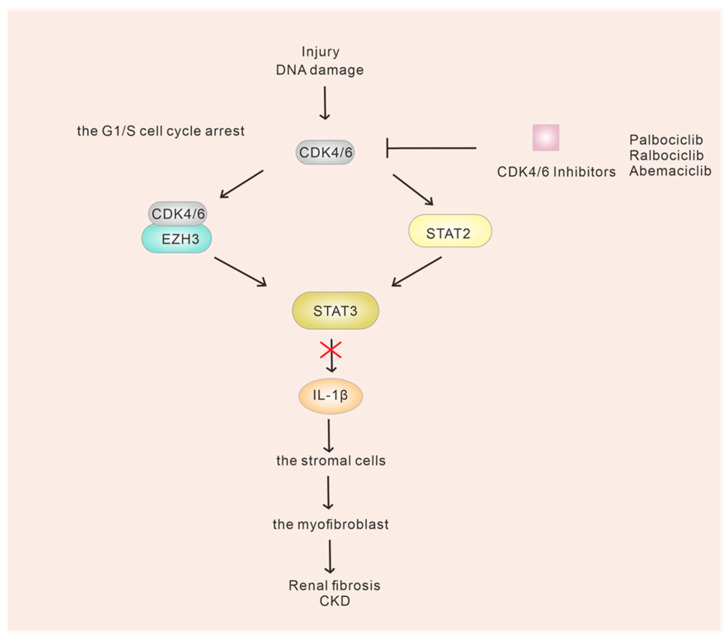
Model diagram of the effect of CDK4/6 inhibitors on CKD. In chronic kidney injury, CDK4/6 may induce STAT3 activation either through the methyltransferase EZH3 or by directly binding to the STAT2 and possibly block cell cycle progression in G1/S phase through a pathway involving STAT3/IL-1β. But, the precise mechanism by which CDK4/6 initiates STAT3 activation remains unclear.

**Figure 6 ijms-24-13558-f006:**
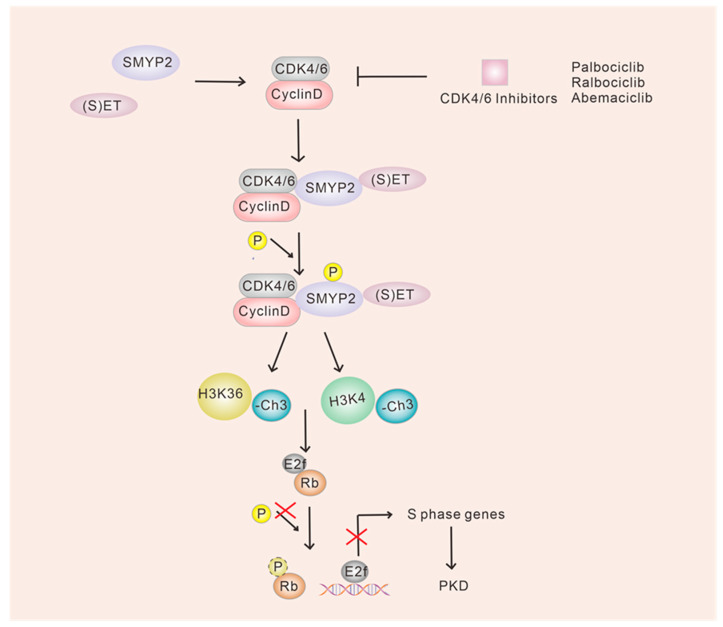
Model diagram of the effect of CDK4/6 inhibitors on PDK. SMYD2 is the new substrate of CDK4/6, which exhibits overexpression in renal tissues and cell lines derived from mice with autosomal dominant PKD. The interaction between SMYD2 and CDK4/6 may be dependent on the presence of the (S)ET structural domain. CDK4/6 interacts with SMYD2, resulting in the phosphorylation and activation of SMYD2. This activation leads to SMYD2-mediated methylation of histones, including histone H3 lysine 4 (H3K4) and histone H3 lysine 36 (H3K36), thereby influencing S gene transcription. However, much research is still needed to investigate and confirm the mechanism of action of CDK4/6 inhibitors in the treatment of PDK.

## Data Availability

Not applicable.

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
