# Peer review of "The Therapeutic Potential of CDK4/6 Inhibitors, Novel Cancer Drugs, in Kidney Diseases"

_ijms, 2023, doi:10.3390/ijms241713558_

Round 1

Reviewer 1 Report

This review contains interesting information on possible therapeutic uses of CDK4/6 inhibitors for kidney diseases.
In my opinion, the title should omit to mention cancer. As a consequence, the part concerning treatment of cancers could be reduced since this aspect is now well documented and discussed in many recent reviews.
However, the necessary caution in using CDK4/6 inhibitors should be stressed, also in treating cancers. Actually, it would be  worthwhile discussing the recent paper by Maissoune Hajir et al. (Current Problems in Cancer: Case Reports, Vol 11 2023) that reports on ribociclib-induced acute kidney injury in patients with advanced breast cancers.
Several repetitions concerning the present model of action of CDK4/6 inhibitors are present in the text and should be avoided.

I did my best to identify the most evident flaws in the text as indicated in the following list. However, a critical revision of the text should be performed.

Page 2
line 62 A CDK
ligne 66 transcription
ligne70 CDK-cyclins compl ?
lines 77-80, unclear, to be reformulated
89, not « The »
Page 3
line 121 unclear, to be reformulated
Page 7
line 243-244 delete eventually
line 246, delete endocrine
line 263, replace belongs to with participates in
line 279 delete can be confirmed
line 293-294 unclear, to be reformulated
page9
line 338 delete However
page 11
line 406  reference 138
line 416 Figure 6
page 13
line 478 delete endocrine

I did my best to identify the most evident flaws in the text as indicated in the following list. However, a critical revision of the text should be performed.

Page 2
line 62 A CDK
ligne 66 transcription
ligne70 CDK-cyclins compl ?
lines 77-80, unclear, to be reformulated
89, not « The »
Page 3
line 121 unclear, to be reformulated
Page 7
line 243-244 delete eventually
line 246, delete endocrine
line 263, replace belongs to with participates in
line 279 delete can be confirmed
line 293-294 unclear, to be reformulated
page9
line 338 delete However
page 11
line 406  reference 138
line 416 Figure 6
page 13
line 478 delete endocrine

Author Response

Comments and Suggestions for Authors

This review contains interesting information on possible therapeutic uses of CDK4/6 inhibitors for kidney diseases.

In my opinion, the title should omit to mention cancer. As a consequence, the part concerning treatment of cancers could be reduced since this aspect is now well documented and discussed in many recent reviews.

Response: Thank you for your valuable comments and suggestions on our manuscript. We appreciate your feedback and agree with your suggestion to revise the title of our article. We agree that the mention of cancer in the title might be misleading and not accurately reflect the focus of our study, which is the therapeutic potential of CDK4/6 inhibitors for kidney diseases.

In light of this, we will modify the title to better align with the content of our manuscript. We will also take into consideration your suggestion to reduce the section discussing the treatment of cancers, as this aspect has been extensively covered in recent reviews. Our revised manuscript will instead focus more on the potential therapeutic applications of CDK4/6 inhibitors specifically for kidney diseases.

Once again, we appreciate your insightful comments, and we believe that these revisions will significantly improve the clarity and relevance of our manuscript.

However, the necessary caution in using CDK4/6 inhibitors should be stressed, also in treating cancers. Actually, it would be worthwhile discussing the recent paper by Maissoune Hajir et al. (Current Problems in Cancer: Case Reports, Vol 11 2023) that reports on ribociclib-induced acute kidney injury in patients with advanced breast cancers.

Several repetitions concerning the present model of action of CDK4/6 inhibitors are present in the text and should be avoided.

I did my best to identify the most evident flaws in the text as indicated in the following list. However, a critical revision of the text should be performed.

Page 2

line 62 A CDK

ligne 66 transcription

ligne70 CDK-cyclins compl ?

lines 77-80, unclear, to be reformulated

89, not « The »

Page 3

line 121 unclear, to be reformulated

Page 7

line 243-244 delete eventually

line 246, delete endocrine

line 263, replace belongs to with participates in

line 279 delete can be confirmed

line 293-294 unclear, to be reformulated

page9

line 338 delete However

page 11

line 406  reference 138

line 416 Figure 6

page 13

line 478 delete endocrine

Response: Thank you for your thorough review of our manuscript. We appreciate your suggestion to emphasize the caution in using CDK4/6 inhibitors, both in treating cancers and kidney diseases. We will include a discussion on the recent paper by Maissoune Hajir et al. (Current Problems in Cancer: Case Reports, Vol 11 2023) that highlights ribociclib-induced acute kidney injury in patients with advanced breast cancers.

We acknowledge the repetitions concerning the present model of action of CDK4/6 inhibitors and will ensure they are eliminated in the revised version of the manuscript. We will perform a critical revision of the entire text to address any other flaws identified.

Thank you for your valuable input, and we are confident that these revisions will greatly enhance the quality of our manuscript.

Reviewer 2 Report

Summary:

 The article titled "Therapeutic potential of CDK4/6 inhibitors in cancers and Kidney Diseases” reviews the potential therapeutic role of CDK4/6 inhibitors in various renal diseases beyond their established use in ER+ breast cancer treatment. After an up-to-date review of the CDK4/6-Rb axis, as well as a summary of the use of CDK4/6 inhibitors in ER+ breast cancer patients, the review discusses the mechanisms underlying the effects of CDK4/6 inhibitors in conditions such as acute kidney injury (AKI), chronic kidney disease (CKD), polycystic kidney disease (PKD), and nephritis.

The review poses that CDK4/6 inhibitors may protect against AKI by delaying the G1/S phase transition, potentially involving ATM/ATR-mediated DNA damage responses. In CKD, CDK4/6 inhibitors may slow cell cycle progression, reduce fibrosis, and interact with STAT3 activation, but the complexities of CKD's cell cycle regulation require further exploration. CDK4/6 inhibitors show potential in PKD treatment by targeting SMYD2 (indirectly or directly through Rb) and potentially restoring primary cilia, offering a new direction for PKD therapy. CDK2 inhibitors' potential benefits in inhibiting cell proliferation in nephritis are briefly mentioned, however, CDK4/6 inhibitors are proposed as a safer alternative due to having fewer side effects.

The article suggests that CDK4/6 inhibitors hold promise for the treatment of various renal diseases. However, the evidence presented is primarily from preclinical studies, animal models, and in vitro experiments. Further clinical research is needed to confirm CDK4/6 inhibitors’ efficacy and safety in treating renal diseases.

General Comments:

This manuscript has no significant weaknesses. The writing is clear, the content is well organized, and the manuscript is suitable for publication in the International Journal of Molecular Sciences.

   - The title accurately reflects the content, however, the capitalization of Kidney Diseases in the title is not necessary.

   - The abstract concisely summarizes the key points and significance of the review.

   - The introduction provides a clear rationale for the review.

   - The literature review is comprehensive.

   - Figures are relevant and clear

   - The review effectively analyzes and synthesizes existing research.

   - The findings are appropriately interpreted and align with the reviewed literature.

   - Limitations are acknowledged and discussed.

   - The conclusion effectively summarizes the main findings.

   - It addresses the review's objectives and implications.

   - The references are current, relevant, and properly cited.

   - There are no missing or inaccurate references.

    - The writing is clear, concise, and well-structured.

   - Aside from capitalization in the title, there are no obvious grammatical errors that need correction.

 Recommendation:

- The manuscript is suitable for publication.

OK

Author Response

Comments and Suggestions for Authors

Summary:

The article titled "Therapeutic potential of CDK4/6 inhibitors in cancers and Kidney Diseases” reviews the potential therapeutic role of CDK4/6 inhibitors in various renal diseases beyond their established use in ER+ breast cancer treatment. After an up-to-date review of the CDK4/6-Rb axis, as well as a summary of the use of CDK4/6 inhibitors in ER+ breast cancer patients, the review discusses the mechanisms underlying the effects of CDK4/6 inhibitors in conditions such as acute kidney injury (AKI), chronic kidney disease (CKD), polycystic kidney disease (PKD), and nephritis.

The review poses that CDK4/6 inhibitors may protect against AKI by delaying the G1/S phase transition, potentially involving ATM/ATR-mediated DNA damage responses. In CKD, CDK4/6 inhibitors may slow cell cycle progression, reduce fibrosis, and interact with STAT3 activation, but the complexities of CKD's cell cycle regulation require further exploration. CDK4/6 inhibitors show potential in PKD treatment by targeting SMYD2 (indirectly or directly through Rb) and potentially restoring primary cilia, offering a new direction for PKD therapy. CDK2 inhibitors' potential benefits in inhibiting cell proliferation in nephritis are briefly mentioned, however, CDK4/6 inhibitors are proposed as a safer alternative due to having fewer side effects.

The article suggests that CDK4/6 inhibitors hold promise for the treatment of various renal diseases. However, the evidence presented is primarily from preclinical studies, animal models, and in vitro experiments. Further clinical research is needed to confirm CDK4/6 inhibitors’ efficacy and safety in treating renal diseases.

General Comments:

This manuscript has no significant weaknesses. The writing is clear, the content is well organized, and the manuscript is suitable for publication in the International Journal of Molecular Sciences.

The title accurately reflects the content, however, the capitalization of Kidney Diseases in the title is not necessary.

Response: We appreciate your positive feedback on our manuscript. We will make the necessary adjustments to the title and remove the capitalization of Kidney Diseases.

The abstract concisely summarizes the key points and significance of the review.

The introduction provides a clear rationale for the review.

The literature review is comprehensive.

Figures are relevant and clear

The review effectively analyzes and synthesizes existing research.

The findings are appropriately interpreted and align with the reviewed literature.

Limitations are acknowledged and discussed.

The conclusion effectively summarizes the main findings.

It addresses the review's objectives and implications.

The references are current, relevant, and properly cited.

There are no missing or inaccurate references.

The writing is clear, concise, and well-structured.

Aside from capitalization in the title, there are no obvious grammatical errors that need correction.

Recommendation:

The manuscript is suitable for publication.

Comments on the Quality of English Language

OK

Round 2

Reviewer 1 Report

This review has been extensively and correctly revised.